# Decision-Making Mechanism of Farmers in Land Transfer Processes Based on Sustainable Livelihood Analysis Framework: A Study in Rural China

Hongbin Liu [1], Hebin Zhang [1], Yuxuan Xu [2] and Ying Xue [2,*]

1 College of Land and Environment, Shenyang Agricultural University, Shenyang 110866, China; liuhongbinsy@syau.edu.cn (H.L.); 2023220489@stu.syau.edu.cn (H.Z.)
2 College of Economics and Management, Shenyang Agricultural University, Shenyang 110866, China; 2020200181@stu.syau.edu.cn
* Correspondence: xueying@syau.edu.cn

**Abstract:** The act of land transfer in rural areas is an important decision-making mechanism for farmers, to enhance resource allocation efficiency and promote capital mobility, and this act is of strategic importance in promoting the level of agricultural scale and mechanization, land system change, and, thus, the sustainable development of livelihoods and production in China. This study aims to explore farmers' decision-making mechanisms in the process of land transfer in rural areas, by constructing a theoretical framework. Structural equation modeling was used, based on data from a survey of rural families in the Liaoning Province area of Northeastern China. The main findings are as follows: (1) The seven types of exogenous latent variables, including environmental vulnerability, policy, and five main livelihood assets (natural capital, physical capital, financial capital, human capital, and social capital), are intermediary in farmers' land transfer behavior, which then positively affect farmers' livelihood outcomes. (2) Among the exogenous latent variables affecting farmers' land transfer, human assets have the most significant positive effect, followed by social assets and physical assets, family labor force share, frequency of work information exchange, and number of production tools, greatly affect the corresponding variable. (3) Natural assets play the most important role and have a negative effect on farmers' land transfer decision; contracted area of land per family is the greatest impacted measurable variable of this. The results of the study suggest that the government should strengthen skills training for farmers, improve the land transfer policy system, and provide appropriate subsidies in a regionally targeted manner. Thus, it can promote the transformation of Chinese-style agricultural modernization and achieve rural revitalization.

**Keywords:** farmers; sustainable livelihood analysis framework; land transfer behavior; structural equation model

## 1. Introduction

As a fundamental industry in the country's economic development, the level of agricultural development influences the floating situation of various industries within the market mechanism. The sustainable development of agriculture is the key to guarantee social welfare and maintain political stability [1–5]. However, China's agricultural production model is still dominated by small-scale farms operated by individuals. Land has always been the most stable and sustainable livelihood guarantee for farmers. It not only has economic value, but also has non-economic value. Due to a long institutional evolution, complex natural conditions, and traditional human characteristics, the "land fragmented" mode of operation has gradually revealed more disadvantages, such as higher production costs, lower production efficiency, and a slow improvement of farmers' welfare [6–9]. Meanwhile, as the trend of urban–rural integration evolves, the surplus agricultural labor force is shifting to the non-agricultural sector, which offers a higher "pay rate" [10,11]. This change

in the structure of human capital has stimulated the need for resource reallocation and consolidation and has promoted the transfer of land resource use rights [12,13]. Land transfer, as an effective measure to expand the scale of agricultural operations, improve agricultural productivity and operational efficiency, and modernize agricultural development, is also gradually receiving a great deal of attention from the Chinese government [14–16].

To this end, in December 2014, the seventh session of the 13th National People's Congress (NPC) voted on a decision to amend the law on rural land contracting, which will legalize the system of "separation of the three rights" of collective ownership of land, contractual rights of farmers, and land management rights in rural land, with a focus on liberalizing land management rights [17,18]. Since then, the national government has continued to introduce improvements to the relevant land system and is committed to promoting the transfer of land management rights and the appropriate scale and intensification of land management [19,20]. The area of rural land transferred reached 532 million mu in 2020, accounting for 34.08% of the total arable land under family contract [21]. Nevertheless, as rational economic persons, farmers will consider the family's capital as a whole before making land transfer decisions and will choose actions that are more beneficial to them, based on the richness of their labor, land, technology, and other livelihood capital [22–24]. The land transfer situation in China still suffers from a series of problems, such as small business scale, lack of motivation, etc. [25–27]. While farmers are the micro subjects of agricultural economic behavior and the decision-makers of land use behavior, changes in the land transfer situation in rural areas depend on farmers' livelihood decisions [28–30]. Therefore, how to guide farmers to transfer their land in a rational and regulated manner on the basis of maintaining the sustainability of farmers' livelihoods has become the focus and difficulty of improving the efficiency of resource utilization.

By combing through the relevant literature, it is easy to find that current research on farmers' land transfer behavior mainly focuses on the scale of land transfer [16], land transfer mode [14], land transfer system reform [31], land transfer willingness and behavior [32–35], factors influencing land transfer behavior [36–38], land transfer performance [31,39], etc. Through analysis, it can be found that scholars have carried out richer research on land transfer behavior, which clearly shows that land transfer has changed the relationship between farmers and land. In particular, by promoting the flow and concentration of urban and rural resources, it provided opportunities for farmers to diversify their livelihood choices and provided a source of motivation for the recapitalization of farmers' livelihoods [40–42]. These research results provide a rich theoretical foundation for this study, but there are still several shortcomings. First, in terms of research perspective, although some scholars have started to step away from the influence of multidimensional livelihood assets on farmers' land transfer behavior, there are few studies that systematically study external vulnerability as a variable. Second, in terms of research content, the influence of livelihood capital on farmers' land transfer and its underlying mechanism of action are yet to be clarified. Third, most of the existing research literature on land transfer is limited to the use of logistic and probit models to study farmers' willingness and actual behavior. However, this measurement method has only been applied in a single mode and requires further optimization. As farmers' land transfer behavior involves a complex decision-making process, new methods should be explored to validate it.

In summary, livelihood sustainability, as an important factor in land transfer transactions, has a significant impact on the livelihood decisions of a limited group of rational farmers. Farmers' livelihood strategies play an important role in improving their economies of scale, productivity, and long-term household welfare; it is a key issue to clarify the mechanisms influencing farmers' participation in land transfer within a sustainable livelihood framework. Based on the framework of sustainable livelihood analysis, constructing an analytical framework for understanding the decision-making mechanism of farmers' land transfer behavior from the perspective of sustainable livelihoods has been developed. Using a structural equation model, the theoretical framework was empirically tested using data from a survey of 777 farming households in Liaoning Province. The study may have

two contributions. Theoretically, it can more accurately show the decision-making process of small-scale farmers' land transfer behavior and provide new research ideas, analytical frameworks, and methodological systems for the application of the sustainable livelihood analysis framework and the thesis of how to promote the development of agricultural scale and intensification. On a practical level, this study can provide suggestions at the micro level, for guiding farmers to participate in land transfer and to promote the improvement of the rural land exchange market. On a macro level, this study can provide an empirical basis for promoting the process of agricultural modernization, enhancing the level of agricultural mechanization and production efficiency, improving the welfare of farmers, and maintaining national food security.

## 2. Materials and Methods

### 2.1. Research Area

In order to ensure the typicality and representativeness of the data, the empirical analysis is based on field survey data from Liaoning province (118°53′~125°46′ E, 38°43′~43°43′ N) in Northeastern China, while the study area is selected from Sujiatun District in Shenyang city (41°11′~43°2′ N, 122°25′~123°48′ E) in central Liaoning province and Donggang city under Dandong city (123°22′~125°42′ E, 39°43′~41°09′ N) in southeastern Liaoning province (see Figure 1). The overall topography of Sujiatun District is not high, with mainly plain arable land, supplemented by low hills and mountains, with the highest elevation being 312 m. The northern part of Donggang City is undulating, with significant differences between the north and the south, while the coastal area is dominated by plains. Both regions have a temperate monsoon climate zone with simultaneous rain and heat and suitable temperatures for agriculture [43,44]. The reasons for selecting these two regions as the study area include the following aspects. First, the heterogeneity of the socio-economic environment. Sujiatun District is a suburban area close to Shenyang city and, according to Tu Neng's theory of agricultural location, the development of this area belongs to the urban agricultural area, which has a high level of economic development and promising market prospects [45]. According to data from the Sujiatun Bureau of Statistics, the GDP was CNY 22,715.45 million in 2019 and the total value of agricultural production accounted for 6.78%; the total population was 425,000 and the rural population accounted for 14.21%. In contrast, Donggang, which is adjacent to the Yellow Sea and Yalu River and located in the border area between China and North Korea, is the largest traditional rice growing area in Liaoning Province and, in recent years, has also developed strawberry farming and other special agriculture, based on the advantages of its sea and land gateway, which is considered a key area for agricultural industry transformation and upgrading [46]. According to data from the Donggang Bureau of Statistics, the total regional production was CNY 21,284.51 million in 2019 and the total value of agricultural production accounted for 35.78%; the total population was 592,248 and rural population accounted for 73.78%. It is typical and representative to explore farmers' livelihood strategies in this region. Second, the heterogeneity of the natural regional environment. The two agricultural regions selected for this study have a diversity of topographic and geomorphological characteristics. Sujiatun District is located on the transition zone between the Liaodong hills and the Liaohe Plain, with low mountainous areas in the east, hilly areas in the middle, and plains in the west, with moderate overall topographic undulations. Donggang City has distinctive topographic features, with a high north and a low south, and the landform is distributed in a stepped pattern from the low mountainous hilly area to the receding sea plain. Survey activities based on this area can better clarify the ideas of land transfer development level improvement and can help to mobilize farmers' enthusiasm to promote agricultural-scale development. Third, the two study areas are located in the core area of black soil protection in Northeast China. The relevant research results have very important practical significance for promoting the moderate-scale management of land resources in the black soil area of Northeast China, improving the utilization efficiency of cultivated land resources and achieving a connection between small farmers and modern agriculture. At the same

time, it also provides a valuable reference for international researchers conducting relevant research in case analysis [47–49].

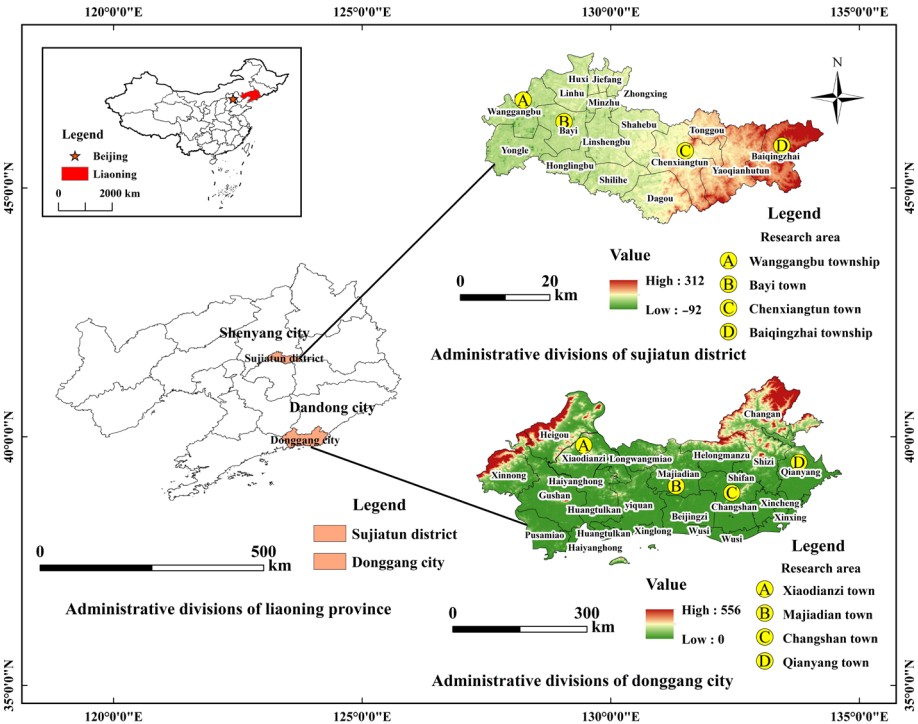

**Figure 1.** Study area map and spatial distribution of sample village.

## 2.2. Data Source

To test the sustainable livelihoods analysis framework, this study chose to use a multi-level stratified sampling method to determine the specific research sites in the design of the research project between February and March in 2018. First, the topographical characteristics, distance from the provincial capital, and economic level difference status among regions were fully considered and Sujiatun District of Shenyang City and Dong-gang City under Dandong City were selected as the study areas. Second, four of the most representative townships were selected for each county, taking into account the township characteristics (shown in Table 1); as a result, eight townships were selected. Third, follow-ing the principle of random sampling, two to four natural villages were selected in each township and the data of natural villages in each township depended on the land area and population area of the township; as a result, 20 villages were chosen.

**Table 1.** The characteristics of townships.

| Townships | Distance from the County (km) | Population (People) | Area of Farm Land (hm²) |
|---|---|---|---|
| Wanggangpu | 18 | 17,328 | 2866 |
| Bayi | 24 | 18,150 | 3403 |
| Chenxiangtun | 45 | 17,052 | 3402 |
| Baiqingzhai | 55 | 11,941 | 6720 |
| Xiaodianzi | 49 | 23,131 | 5530 |
| Majiadian | 30 | 32,006 | 5250 |
| Changshan | 12 | 47,268 | 8666 |
| Qianyang | 10 | 60,684 | 7089 |

Data sources: Sujiatun and Donggang Bureau of Statistics.

The types of farmers in the survey area were categorized as rented-out, rented-in, and non-transferred. A problem of underestimating the sample of renters tends to occur in

random surveys of farmer samples, because some renters may permanently or temporarily migrate to other locations, making it impossible to find that type of farmer during the walk-in survey. To reduce this type of bias, the proportion of farmers in each category in the village was first estimated through interviews with village officials. This estimate was then used to adjust the number of farmers in each township visited. This sampling strategy was able to ensure that the proportions of the three types of rural families (rented-out, rented-in, and non-transferred) was consistent with the overall proportion of the township. Next, we randomly selected 30–50 farmers in each village. The study used an in-home sampling format, in which researchers who had undergone rigorous training and study in the early stages went deep into farmers' families and fields to conduct sampling; the entire survey process adhered to the principle of rigor. The survey obtained data from 811 farmers in 20 villages across 8 towns. After collecting, sorting, and screening the data, eliminating abnormal data and missing value data, a total of 777 valid data were obtained and the questionnaire efficiency reached 95.80%. Overall, the sample distribution is reasonable and can reflect the basic situation of farmers' participation in land transfer; the basic characteristics of the sample are shown in Table 2.

**Table 2.** Basic characteristics of the sample.

| Variable | Indicator | Frequency | Percentage | Variable | Indicator | Frequency | Percentage |
|---|---|---|---|---|---|---|---|
| Gender | Male | 726 | 93.4% | Land Transfer | Land transferred in | 204 | 26.3% |
| | Female | 51 | 6.6% | | Land not transferred | 359 | 46.2% |
| Academic qualifications | Illiterate | 22 | 2.8% | | Land transferred out | 214 | 27.5% |
| | Elementary | 263 | 33.8% | Address | Sujiatun District | 378 | 48.6% |
| | Middle | 413 | 53.2% | | | | |
| | High | 70 | 9.0% | | Donggang City | 399 | 51.4% |
| | Bachelor's degree | 9 | 1.2% | | | | |
| Whether to work outside | Yes | 263 | 33.84% | Skills training | Yes | 219 | 28.18% |
| | No | 514 | 66.15% | | No | 558 | 71.81% |

*2.3. Research Methodology*

2.3.1. Theoretical Analysis

The livelihood of farmers has been a topic of concern in various countries, regions, and academic circles [50–53]. Early studies on livelihood primarily centered on poverty, specifically on income levels, consumption capacity, and other factors related to basic living needs [54]. As research has progressed, and with the advancement of poverty alleviation practices and theoretical development, it is recognized that income and consumption are no longer the sole indicators to assess poverty [55]. The UK Department for International Development (DFID) proposed a sustainable livelihood analysis (SLA) framework, which consists of five main components, namely, vulnerability context, five types of livelihood capital, structural and institutional shifts, livelihood strategies, and livelihood outputs, which are interlinked and influence each other [56–58]. Among them, livelihood capital is the central aspect of the sustainability analysis framework and includes the following five components: natural capital, physical capital, financial capital, human capital, and social capital, which influence the formation of livelihood strategies to different degrees. The research conducted by applying the sustainability analysis framework has extensively covered many cross-cutting disciplines such as tourism industry, macroeconomic analysis, social infrastructure, and migration issues [59–62].

On this basis, in this study, the sustainable livelihoods framework of DFID was slightly adjusted, by combining it with the direction and area of land transfer to create a new framework, as illustrated in Figure 2. The paper specifically examines decision-making mechanism of farmers' land transfer behavior, introducing a new solid line arrow denoting 'livelihood assets → livelihood strategies → livelihood outcomes'. Furthermore, it is noted that environmental vulnerabilities and policies can indirectly influence livelihood strategies through livelihood assets and may also have a direct impact on farmers' livelihood strategies.

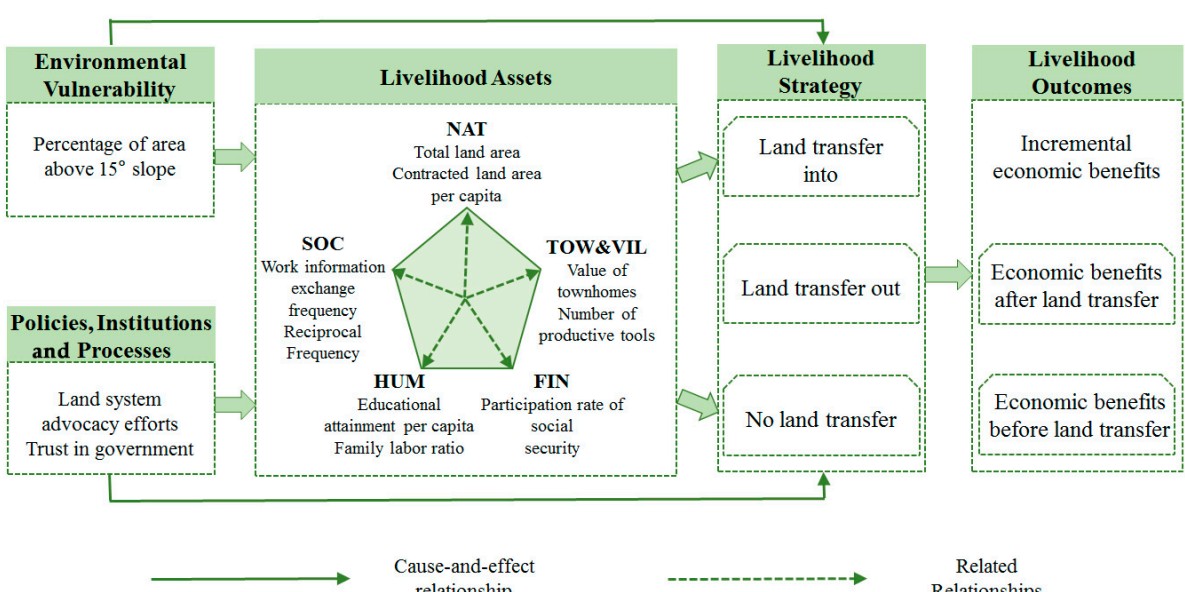

**Figure 2.** The analytical framework for the decision-making mechanism of farmers' land transfer behavior, from the perspective of sustainable livelihoods.

(1) The influence mechanism of environmental vulnerabilities and policies to farmers' land transfer behavior

The external environment is an important factor for farmers to consider when making livelihood strategy choices based on their livelihood assets, and this environmental factor also affects farmers' access to livelihood assets, including both environmental/contextual vulnerability and policies, institutions and processes. Environmental/contextual vulnerability includes trends, shocks, and seasonal fluctuations in the natural and socioeconomic environment, and the more pronounced a farmer's environmental/contextual vulnerability is, the more likely he or she is to choose not to transfer land. Policies, institutions, and processes refer to the policies formulated or promoted by the government, the management level of the government and institutions, and the factors in the implementation process. In the decision mechanism model of farmers' land transfer behavior, the laws, regulations, and policies of the land transfer system and their implementation affect farmers' access to livelihood assets and farmers' enthusiasm for land transfer; in general, the better the policies, institutions, and their processes, the more likely farmers are to transfer their land. Based on their livelihood assets, farmers need to choose appropriate livelihood strategies in response to external environmental shocks, in order to achieve sustainable livelihood outcomes.

(2) The influence mechanism of five major livelihood assets to farmers' land transfer behavior

In the SLA framework, livelihood assets are the basis for farmers to make decisions on livelihood strategy choices. Natural assets refer to the stock of natural resources, including land, water resources, forest products, etc. For the issue of land transfer, farmers' land assets are mainly considered and, generally speaking, the more natural assets farmers have, the more they tend to transfer to land. Material assets are infrastructure and production tools including transportation facilities, dwellings, production equipment, etc., which influence the choice of livelihood strategies by improving production efficiency. Human assets are important assets for farmers to choose their livelihood strategies and are one of the most important factors of production in the process of agricultural modernization, including education, knowledge and skills, and labor capacity, etc. Generally speaking, the more human assets, the more they tend to be transferred out of the land. Financial assets are financial flows that have an impact on production, including income and available social security, which can be transformed into other assets, and are, therefore, one of the

most important assets affecting farmers' livelihoods. Social assets are social resources that farmers can use, including the sum of a series of social relationships. Farmers can bring valuable assets to their families through access to effective information, and farmers with more social assets are more likely to have opportunities to engage in other production, rather than just agricultural production; therefore, the more social assets they have, the more they tend to transfer out of land. Livelihood strategy is the way farmers adjust, combine, and use their livelihood assets in order to maintain the sustainability of their livelihoods and it usually contributes positively to farmers' livelihood outcomes. Livelihood outcomes are the outputs of improved living standards and optimal resource use that result from farmers' livelihood strategy choices, thus enabling farmers to maintain the sustainability of their livelihood development.

### 2.3.2. Structural Equation Model

In order to validate the theoretical structure of "livelihood capital–livelihood strategy–livelihood outcome" within the framework of sustainable livelihood analysis, this study uses structural equation modeling (SEM) for multivariate statistical testing. The SEM model was introduced by Joreskog and Wiley in the 1970s as a multivariate statistical analysis method to analyze the interaction between variables, theoretical model testing, impact path analysis, and sample data fitting effects [63]. The model has the advantages of multi-factor analysis, high error accommodation, and fitting effect analysis and it is more suitable for this study [64,65]. First, previous econometric models are mostly limited to single-factor analysis; SEM breaks this constraint and can analyze and measure multiple dependent variables and factor structures as a way to clarify the complex relationships among the factors [66,67]. Second, in the face of latent variables that are difficult to measure accurately with indicators, SEM models allow for a greater degree of measurement error and prevent sample bias due to constraints such as observation difficulty [68]. The application of the model can ensure scientific and precise analysis results. Third, the model can report on the degree of fit of the data, which, in turn, facilitates the judgment of the validity of the model. Existing studies have used SEM models to address the impact mechanisms and path analysis in various areas [69,70]; for example, Pham used the method to examine the influence of different dimensions of HRM on the competitive advantage and performance levels of firms [71]. Mulyaningsih et al. then used this method to analyze the significant role played by strategic planning in the process of competitive advantage formation for small and medium-sized firms in Indonesia [72]. Therefore, this study adopts this model to systematically analyze the influencing factors in the decision-making process of farmers' land transfer behavior, so as to investigate how farmers maintain the sustainability of family livelihoods through the choice of land transfer as a livelihood strategy. At the same time, for the two groups of farmers, the mechanisms of decision-making are different, to a certain extent. In order to better compare the influence paths of each dimension on farmers' decision-making behavior in the two forms, two SEM models are constructed in this study to compare and analyze in the empirical test. The SEM base model constructed in this paper is as follows:

$$\eta = B\eta + \Gamma\xi + \zeta \tag{1}$$

$$\mathrm{x} = \Lambda_x\xi + \delta \tag{2}$$

$$\mathrm{y} = \Lambda_y\eta + \varepsilon \tag{3}$$

Equation (1) is a structural model to describe the linear relationship between latent variables, where $\xi$ is the exogenous latent variable, which, in this study, refers to natural assets, physical assets (urban), physical assets (rural), human assets, financial assets, social assets, environmental/contextual vulnerability, policies, institutions, and processes. $\eta$ is the endogenous latent variable, which, in this study, refers to natural assets, livelihood strategies, and livelihood outcomes. B denotes the matrix of effect coefficients of endogenous latent variables on endogenous latent variables, $\Gamma$ denotes the matrix of effect coefficients of exogenous latent variables on endogenous latent variables, and $\zeta$ denotes

the vector composed of residual terms. Equations (2) and (3) are measurement models used to describe the linear relationship between latent and measurable variables. $\Lambda x$ and $\Lambda y$ denote the regression coefficients of the measurable variables on the exogenous latent and endogenous latent variables, respectively; x and y subscale the exogenous measurable and endogenous measurable variables; and $\delta$ and $\varepsilon$ denote the measurement errors of the measurable variables on x and y, respectively.

The detailed structural paths and underlying path assumptions are presented in Table 3. It is of interest to note that, for the objectives of the study, the selection of indicators for external environmental vulnerability and policy variables in this study revolves around land as a natural resource. Thus, although external factors stimulate changes in the initial level of farmers' livelihood assets, the stimulus that emerges in this study acts to a greater extent on farmers' natural assets, due to the more naturally constrained nature of land transfer behavior. Natural assets are extremely environmentally constrained and policy-directed and this process of farmers' access to natural assets can significantly change under the intervention of external factors. The lower the ecological vulnerability, the lower the risk of acquiring and holding natural assets and the weaker the constraint. In this case, farmers will increase the level of natural assets; whether the policy orientation is supportive or inhibitory plays an important role in farmers' risk prediction. The stronger the policy orientation is, the more abundant the farmers' natural assets. Among all the livelihood assets in this study, the influence of external factors on natural assets received more attention.

**Table 3.** Model variables and their descriptions.

| Latent Variable | Observed Variables | Definition | Min | Max | Mean | S.D. |
|---|---|---|---|---|---|---|
| ENV | Percentage of area with slope above 15° in the region | Area of slope above 15°/total area of area | 0.00 | 13.19 | 2.88 | 4.24 |
| POL | Land system advocacy efforts (pol1) | No strength = 1, low strength = 2, average = 3, high strength = 4, high strength = 5 | 1.00 | 5.00 | 3.55 | 1.30 |
| | Trust in government officials (pol2) | Very low = 1, low = 2, average = 3, high = 4, very high = 5 | 1.00 | 5.00 | 3.38 | 1.38 |
| | Satisfaction with the village committee election system (pol3) | Unsatisfactory = 1, fair = 2, satisfactory = 3 | 1.00 | 3.00 | 2.49 | 0.66 |
| | Support for land transfer by village councils (pol4) | Yes = 1, No = 0 | 0.00 | 1.00 | 0.59 | 0.49 |
| NAT | Total contracted area of family (nat1) | Total arable land contracted by farmers' families | 0.00 | 70.00 | 11.20 | 5.95 |
| | Contracted area of land per family (nat2) | Family arable land area/number of people | 0.00 | 23.30 | 3.32 | 2.17 |
| | Average size of a single plot (nat3) | Family arable land area/number of plots | 0.00 | 35.00 | 4.87 | 4.31 |
| TOW | Townhouse Values | None = 0; ≤200,000 = 1; 21–400,000 = 2; 41–600,000 = 3; 61–800,000 = 4; >800,000 = 5 | 0.00 | 5.00 | 0.29 | 0.74 |
| VIL | Number of production tools (vil1) | Number of productive tools owned by family | 0.00 | 3.00 | 0.28 | 0.68 |
| | Total value of farm assets (vil2) | None = 0; ≤20,000 = 1; 30–40,000 = 2; 40–60,000 = 3; 60–80,000 = 4; >80,000 = 5 | 0.00 | 5.00 | 0.27 | 0.58 |
| | Rural Housing Value (vil3) | ≤100,000 = 1; 11–15 million = 2; 16–20 million = 3; 21–25 = million = 4; >250,000 = 5 | 1.00 | 5.00 | 1.29 | 0.72 |

**Table 3.** *Cont.*

| Latent Variable | Observed Variables | Definition | Min | Max | Mean | S.D. |
|---|---|---|---|---|---|---|
| FIN | Proportion of family members participating in social security | Number of family participants in social security/total family size | 0.00 | 1.00 | 0.32 | 0.34 |
| HUM | Family education per capita (hum1) | (Number of people in elementary school and below × 0.2 + number of people in middle and high school × 0.6 + number of people in college and above × 1)/Total number of people in the family | 0.20 | 1.00 | 0.58 | 0.31 |
| | Family labor force share (hum2) | (Number of young laborers × 1 + number of older laborers × 0.5)/total family size | 0.00 | 1.00 | 0.60 | 0.25 |
| SOC | Frequency of work information exchange (soc1) | Rarely = 1, Less = 2, Average = 3, More = 4, Many = 5 | 1.00 | 5.00 | 2.15 | 1.26 |
| | Frequency of land transfer information exchange (soc2) | Rarely = 1, Less = 2, Average = 3, More = 4, Many = 5 | 1.00 | 5.00 | 2.15 | 1.20 |
| | Frequency of Village Civil Mutual Aid (soc3) | Rarely = 1, Less = 2, Average = 3, More = 4, Many = 5 | 1.00 | 5.00 | 2.42 | 1.36 |
| STR | Land Transfer Out Model (Model I): Land Transfer | Land transferred out = 1, land not transferred = 0 | 0.00 | 1.00 | 0.37 | 0.48 |
| | Land Transfer Model (Model II): Land transfer | Land transferred in = 1 land not transferred = 0 | 0.00 | 1.00 | 0.36 | 0.48 |
| CON | Change in post-flow income (con) | Substantial decrease = 1, slight decrease = 2, no change = 3, slight increase = 4, substantial increase = 5 | 1.00 | 5.00 | 3.06 | 1.06 |

### 2.3.3. Variable Selection

Based on the theoretical framework of the decision-making mechanism of farmers' land transfer behavior, from the perspective of sustainable livelihoods, this study set up 10 latent variables and 20 measurable variables (Table 3), the specific reasons are as follows:

(1) Environmental/contextual vulnerability (ENV). Based on the soil slope class classification criteria of NBSS and LUP, as well as the topographic characteristics of the study area, this study classified the topography into two classes according to the heterogeneity of slope, by referring to Romshoo et al., and set 15° as the dividing line between gentle and moderate slopes [73]. In this study, the proportion of area above 15° slope in each commune (township) was selected to measure farmers' environmental/contextual vulnerability as an external factor affecting farmers' behavior; environmental vulnerability affects actors' livelihood strategies. Specifically, it is explained that the higher the vulnerability of the environment, the higher the sunk risk of input costs in the agricultural production process and the more farmers will tend to avoid expansive livelihood strategies. At the same time, ecological vulnerability also includes the quality of the soil, where the gradual weakening of the soil organic matter layer leads to a reduction in agricultural output, and the farmers' yield remedies through excessive fertilizer use further aggravate ecological vulnerability. This vicious circle can cause farmers to lower their expected returns from agricultural production, which, in turn, creates a willingness to transfer their land out.

(2) Policies, institutions, and processes (POL). Compared to the influence of the natural environment, social factors have a more pronounced role in interfering with farmers' decision-making [74,75]. In this study, four observed variables, namely, the degrees of rights defense of the land system (pol1), the degree of trust in the government (pol2),

the satisfaction with the village committee election system (pol3), and the degree of village collective support for land transfer (pol4), were selected to measure POL. In terms of the existing policy system, the issuance of land contracting certificates and management rights contracts further clarifies property rights boundaries, reduces the frequency of transaction disputes, and stimulates farmers' willingness to participate in land transfer; in terms of institutional organization, the more farmers trust grassroots units and perceive stronger policy support, the more their consideration of risk issues will be greatly reduced, which, in turn, enhances their willingness to participate in land transfer.

(3) Natural assets (NAT). In this study, three indicators—total contracted family land area (nat1), contracted family land area per capita (nat2), and individual contracted plot area (nat3)—are selected to measure NAT, respectively. Due to the constraints of natural conditions, small-scale farmers are increasingly faced with the problems of having more people and less land, insufficient arable land per capita, and fragmented plots, which make it increasingly difficult for farmers to rely on arable land resources to maintain their livelihood sustainability in daily production and operation processes [76,77]. With the further development of industrialization and urbanization, the transfer of surplus agricultural labor has promoted the integration of rural arable land resources and has accelerated land transfer. Specifically, the larger the per capita arable land area of a family, the larger the area of individual plots; additionally, the larger the total contracted arable land area of a family, the easier it is for farmers to rely on natural assets to form large-scale operations and thus obtain economies of scale. Driven by the benefits, farmers are more inclined to transfer into land.

(4) Material assets (TOW and VIL). Considering the weakening of the urban–rural dual system, this study will measure the material assets of farmers from both urban and rural perspectives. Among them, the selected measure of material assets in towns is the value of urban housing. This is due to the fact that farmers with more material assets in urban areas are more adaptable and integrated and are more likely to leave agricultural production and move to urban areas for work [78]. In this study, rural material assets mainly refer to the total value of productive tools, dwellings, and agricultural assets owned by farmers in rural areas; farmers with more rural material assets tend to stay in rural areas to maximize the benefits of asset utilization. Specifically, the greater the number of productive tools, the higher the value of dwellings; in addition, the higher the total value of agricultural assets, the greater the ability of farmers to engage in modern agricultural scale operations, the greater the willingness to stay in the countryside for a long period of time, and the higher the cost of upfront inputs, which makes farmers more willing to expand their scale and transfer to land.

(5) Financial assets (FIN). In this study, the ratio of the number of participants in social security to the total number of households is selected to measure this indicator. This is because the higher the proportion of people participating in social security within rural households, the better their income security. Such a condition can enhance farmers' resilience to risk, prompting them to further expand their production operations or engage in other occupations, and stimulate land transfer behavior. In 2014, the government merged the new rural social pension insurance and urban residents' social pension insurance, to establish a nationwide unified basic pension insurance system for urban and rural residents, reducing the uneven gap between urban and rural areas. At the same time, with the increase in the participation rate, farmers' livelihood resilience has gradually increased, stimulating farmers to change their livelihood strategies, further promoting modernized agricultural operations, and achieving sustainable agricultural development [79,80].

(6) Human assets (HUM). With regard to human assets, this study uses the education level per capita of the family and the number of family laborers, as a percentage, to measure them [81]. This is because the overall literacy and human capital situation within farmers' families affects both agricultural production and operation and land

use. Farmers with low educational attainment have more difficulty acquiring other skills or engaging in off-farm employment, due to the limitations of their knowledge level, and are, therefore, more likely to shift to the land to continue their livelihoods in skilled agricultural production activities. For the human capital of families, there are two specific cases. The more effective labor within the family, the more farmers can choose to allocate their surplus labor to part-time employment, to increase off-farm income and expand family income channels; on the other hand, they can take advantage of labor to transition to land and expand production scale. In both cases, farmers will consider the transfer of land use rights.

(7) Social assets (SOC). In this study, the frequency of information exchange among village folk for labor (soc1), the frequency of information exchange for land transfer (soc2), and the frequency of occurrence of agricultural mutual aid behavior (soc3) were selected as the variables for measuring SOC. Although the rural coverage of communication base stations has been relatively well established and the penetration rate of modern media technology and communication devices has reached a high level, farmers are still at a disadvantaged position in terms of information access, due to constraints of literacy, learning ability, and base station location [82,83]. The most common form of information circulation in villages is direct communication between people. The more frequently farmers communicate with other villagers or foreign villagers, the more information they can obtain, and the more it will influence farmers' decision-making. The more information about non-farm work and land transfer that is exchanged among villagers, the easier it is to motivate farmers to give up agricultural production and operation. The more frequent the mutual assistance among villagers, the more it will reduce the pressure of agricultural production, which, in turn, will stimulate farmers to expand their scale. The specific variables are set as shown in Table 4, where the measure of whether farmers' land transfer behavior meets the goal of sustaining livelihood sustainability is explained by the increment of income.

**Table 4.** Key indicators for model fitting.

| Type of Land Transfer | Name of the Index | Abbr. | Acceptable Fit Values | Fit Values | Results |
|---|---|---|---|---|---|
| Land transfer out | Root Mean Square Error of Approximation | RMESA | <0.08 | 0.050 | Accept |
| | Goodness-of-fit index | GFI | >0.9 | 0.937 | Accept |
| | Comparative fit index | CFI | >0.9 | 0.953 | Accept |
| | Incremental fit index | IFI | >0.9 | 0.953 | Accept |
| | Tacker–Lewis index | TLI | >0.9 | 0.943 | Accept |
| | Cardinality to Degrees of Freedom Ratio | CMIN/DF | <3 | 2.445 | Accept |
| Land transfer in | Root Mean Square Error of Approximation | RMESA | <0.08 | 0.054 | Accept |
| | Goodness-of-fit index | GFI | >0.9 | 0.931 | Accept |
| | Comparative fit index | CFI | >0.9 | 0.949 | Accept |
| | Incremental fit index | IFI | >0.9 | 0.949 | Accept |
| | Tacker–Lewis index | TLI | >0.9 | 0.938 | Accept |
| | Cardinality to Degrees of Freedom Ratio | CMIN/DF | <3 | 2.637 | Accept |

## 3. Results and Discussion

### 3.1. SEM Model Fit and Suitability Analysis

Based on Amos 24.0 software, this study conducted exploratory modeling of the sample data using the great likelihood method; the model was modified asymptotically to construct the best-fit model. In this paper, the applicability of the model was assessed using absolute fit indices and incremental fit indices, etc., and the results are shown in Table 4, where all fit indices are better than or within acceptable values. These estimates

suggest that the impact mechanisms and transmission paths between natural assets (NAT), human assets (HUM), social assets (SOC), physical assets (VIL and TOW), financial assets (FIN), environmental vulnerability (ENV), policy (POL), land transfer (STR), and livelihood outcomes (CON) are acceptable and that SEM has a good fitting effect. Considering the significance and goodness-of-fit indicators of the main paths of the two models together, the constructed model is considered to satisfy the fitness conditions and can be used for the analysis of SEM.

### 3.2. Policies and Environmental Vulnerabilities Affecting Farmers' Land Transfer

In terms of the externalities influencing farmers' formation of livelihood strategies, environmental vulnerability (ENV) is an important factor that damages farmers' natural assets and inhibits land transfer behavior. Under the analysis of the land transfer out model and the land transfer in model, as shown in Figure 3, the mechanisms of influence due to ENV have great similarity. Specifically, the total effects of ENV → NAT are −0.127 and −0.285, respectively, the total effects of ENV → STR are −0.197 and −0.179, respectively, and the total effects of ENV → CON are −0.059 and −0.148, respectively, indicating that the higher the environmental vulnerability, the higher the degree of resistance to farmers' natural assets, and the more it will discourage farmers from participating in land transfer and weaken the effect of sustainable livelihood maintenance. The higher the vulnerability of the environment to farmers' natural assets, the more it will discourage farmers from participating in land transfer and weaken the effect of sustainable livelihood maintenance. Conversely, policies, institutions, and processes (POL) are key elements that promote the preservation of farmers' natural assets, enhance their motivation to participate in land transfer, and maintain a sustainable livelihood status. Systematically, the more sloping the area, the more likely it is to lead to the fragmentation of arable land and, often, less arable land, which is not conducive to agricultural mechanization and ultimately leads to the accentuation of farmers' livelihood vulnerability. In contrast, a sound policy system (pol1), reliable grassroots organizations (pol2 and pol3), and a good policy implementation process (pol4) will help farmers to perceive state encouragement and support. Consequently, they will be more motivated to put their willingness to transfer into practice.

In addition, it is worth noting that both external environmental vulnerability and policy factors have an impact on farmers' initial level of natural asset ownership. First, environmental vulnerability (ENV) consistently inhibits the level of natural asset holdings, regardless of the flow of farmers' land transfers. For each unit increase in the intensity of environmental vulnerability, farmers' land transfer behavior decreases by 0.127 and 0.285 units (for the transfer-out and transfer-in groups, respectively). This is mainly because environmental vulnerability enhances the riskiness of farming behavior; the more sloping and uneven the arable land is, the more its soil quality decreases significantly under long-term management and it is also not conducive to mechanized operations. This riskiness will directly act on farmers' decision-making processes and is an important factor influencing final behavior and, under strong risk perception, farmers will choose avoidant behavior. Therefore, the more vulnerable the external environment is, the less willing the farmers are to choose to transfer their land. Second, farmers in both groups enhance their willingness to participate in land transfer under the effect of policy factors. For each unit increase in the intensity of policies, institutions, and processes, correspondingly, farmers' land transfer behavior increases by 0.103 and 0.241 units (for the transfer-out and transfer-in groups, respectively). This is because the higher the level of support, policy tilt, and trustworthiness of the government as an organization with strong credibility, the more it will reduce farmers' risk preconceptions and motivate them to join in land transfer.

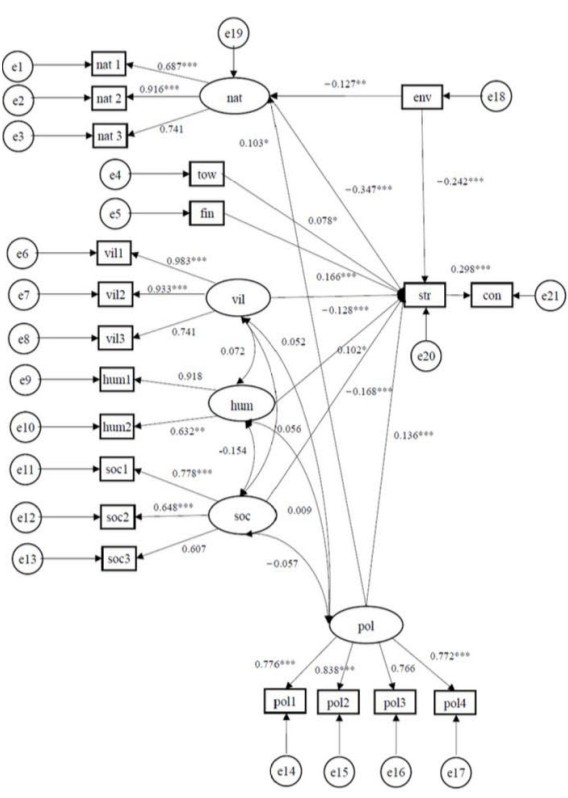

(a) Group of land transfer out

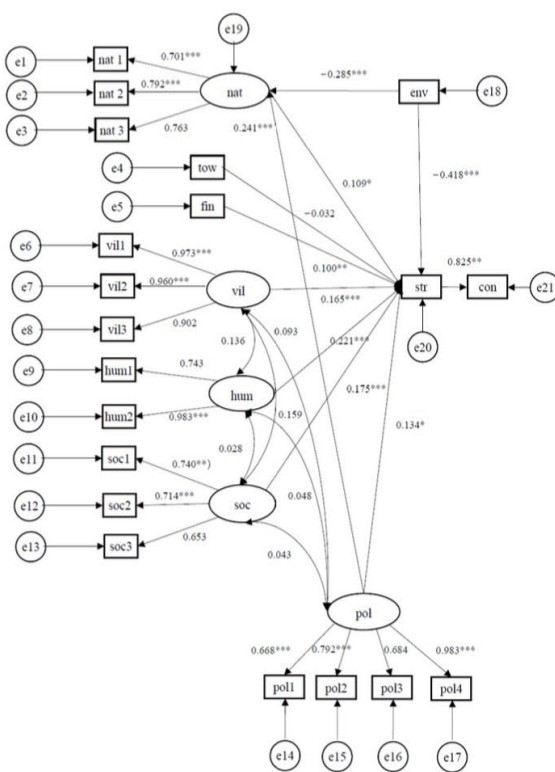

(b) Group of land transfer in

**Figure 3.** Estimation results of the SEM model of farmers' land transfer behavior, where (**a**) is the land transfer out group and (**b**) is the land transfer in group. Note: → the starting end point is the dependent variable and the arrow end point is the outcome variable; *** denotes *p* < 0.001, ** denotes *p* < 0.01, * denotes *p* < 0.05.

### 3.3. *Five Major Livelihood Assets Influencing Farmers' Land Transfer Behavior*

3.3.1. Natural Assets (NAT)

There is an inverse relationship between natural assets (NAT) and farmers' land transfer (STR) behavior, in the case of land use rights transactions with different flow directions. The standardized path coefficient was $-0.347$ ($p < 0.01$) in the case of farmers switching out of land, indicating that NAT was an important "disincentive" for farmers to switch out of land and a major impediment to improving farmers' livelihood outcomes in the later stages of the transaction. The higher the total arable area (nat1) and per capita arable area (nat2) initially contracted by families, and the lower the degree of arable land fragmentation (nat3), the better the natural endowment of farmers, which would significantly "discourage" the emergence of off-farm employment. This is because the more suitable the contracted land is for farming, the more likely it is that farmers tend to keep this part of the land, which leads to the refusal to transfer out of the land and thus has some impact on farmers' livelihood outcomes. For the group of farmers who transferred into the land, NAT, to the extent of $p < 0.10$, motivated farmers to make further livelihood strategies to expand their business. The higher the per capita arable area of the family (nat2) and the lower the individual plot size (nat3), the higher the number of laborers that can be carried, and the higher the number of farmers that tend to continue to shift to farmland when they can sustain their livelihoods through agricultural production. In addition, the lower the degree of fragmentation of arable land, the more favorable it is for large agricultural machinery to carry out mechanized production, whereby further transfer of land will enhance scale efficiency and motivate farmers to moderately expand their production scale.

3.3.2. Physical Assets (TOW and VIL)

Physical assets in urban areas (TOW) have a significant contribution to farmers' behavior of transferring out of land. According to Figure 3, each unit increase in urban assets increases farmers' behavior of transferring out of land by 0.078 units. This indicates that farmers with more physical assets in the town tend to transfer the right to use the rural contracted land. This is mainly because the more material assets farmers have in towns, the more they can secure their survival in the city and the more they tend to engage in non-farm work. In turn, they tend to expand their income and enhance the sustainability of their livelihoods. The effect of the "attraction" exuded by the town on the group of farmers who have transferred their land from others is not obvious, mainly because such farmers have fewer material assets in the town and their livelihood is not enough to support their development in the town. Material assets in rural areas (VIL) "discouraged" farmers' willingness to move out of land, but conversely had a significant effect on "inducing" farmers to move into land ($p < 0.01$). This is because the higher the number of productive tools (vil1), the total value of fixed assets (vil2), and the value of homestead (vil3), the higher the cost of exiting the farming operation; the incremental gains from the later stages of the transaction hardly offset the sunk costs. Conversely, this stronger rural subsistence base will stimulate farmers to further transfer into the land, thus expanding the scale of production and enhancing economic efficiency.

3.3.3. Financial Assets (FIN)

Financial assets (FIN) are a factor that improves farmers' resilience in a virtual form and, according to Figure 3, are among the main factors affecting farmers' land transfer behavior, having a positive effect on all forms of transactions in different streams. For each unit increase in FIN, farmers' behavior in transferring out of land increases by 0.166 units and the significance of this effect is stronger compared to the group of farmers who transfer into land. This indicates that the higher the number of participants in the social security system within the family, the higher the farmers' sense of security, the less constrained their livelihood strategy formulation, and the higher their willingness to participate in land transfer. FIN is the most important factor that promotes the participation of farmers

who transfer out of their land and maintain their livelihood sustainability. This is because, with a secure income for family members, the volatility of gains and losses has less impact on farmers' livelihoods, which can support farmers to participate in the transfer with greater peace of mind. The risk of "loss of land and unemployment" is much less likely to occur after the transfer of farmland, thus ensuring the sustainability of livelihoods. It is noteworthy that the impact of FIN on livelihood strategies and livelihood outcomes is the smallest in the model for farmers who move to land. This may be explained by the fact that the current social security system is not yet strong enough to support the act of moving to land and that arable land takes on the main livelihood security function, as well as the fact that the livelihood situation of farmers who move to land does not improve much, in relation to the rate of participation in social security.

### 3.3.4. Human Assets (HUM)

Human assets (HUM) had the greatest effect on farmers' land transfer behavior, with HUM producing a significant push effect at the 10% and 1% levels in the land transfer out and land transfer in models, respectively. However, this pushiness produces a greater effect for farmers who transfer into land, where each unit increase in farmers' human capital level corresponds to a 0.221 unit increase in land transfer behavior and is the most basic incentive for farmers to purchase land use rights. This indicates that the higher the overall level of literacy knowledge of the family (hum1) and the higher the number of laborers (hum$^2$), the more farmers tend to transfer to land and further develop large-scale operations. According to the process of receiving knowledge, technology, and information, farmers' learning ability is built on the platform of initial literacy. The more knowledge accumulated under the initial conditions, the stronger the ability to filter, absorb, and integrate information at a later stage and the easier it is to make a more rational and scientific livelihood strategy. As for the amount of human capital, as a key factor input in the production process, the greater the amount of labor, the higher the productivity level of farmers. More farmers with such production advantages will improve the efficiency of using advantageous resources, further transfer into the land, expand the scale of operation, and promote the efficiency of agricultural production. The facilitative effect of HUM on livelihood strategies and livelihood outcomes in the land transfer out model is second only to FIN, which indicates that, whether one chooses to engage in large-scale production or work, there is a need for a sufficient labor force with a certain educational literacy.

### 3.3.5. Social Assets (SOC)

From the perspective of SOC, the higher the frequency of exchanging information on labor (soc1), the frequency of exchanging information on transfer (soc2), and the agricultural mutual aid behavior among village people (soc3), the lower the disincentive for farmers who transfer out of their land. This indicates that the greatest degree of risk and uncertainty is associated with outworking under the influence of differential perceived risk pressure. The higher the frequency of exchanging work information among farmers, the more they perceive the pressure of employment, so they will inhibit their land transfer out behavior. In contrast, agricultural mutual aid among village people can greatly reduce their pressure and, when they feel that agricultural cultivation is easier, they are willing to keep part of their arable land to provide themselves with some employment security. However, in the current agricultural modernization process, this traditional farming model is not adapted to the development of agricultural modernization and it is obvious that the transferred families are more adapted to entering the city to work and that retaining farmland for part-time production is not suitable for the sustainability of their livelihoods; this strategy will eventually worsen the livelihood outcomes and lead to a decline in their income levels. The frequency of exchange of information on land transfer (soc2) has the greatest effect on farmers' livelihood strategies for farmers who have transferred to land, suggesting that the more farmers know about land transfer, the easier it is for them to grasp useful information.

The lower the risk pressure the farmers perceive from land transfer, the more likely they are to act accordingly to expand their scale.

### 3.4. Impact of Land Transfer Behavior (STR) on Farmers' Livelihood Outcomes (CON)

Farmers' land transfer behavior as an expression of livelihood strategy is consistent with the sustainable livelihood analysis framework, as shown in Figure 3. Under the influence of livelihood assets—environmental/contextual vulnerability; policies, institutions, and processes—farmers' participation in land transfer as a livelihood strategy (STR) significantly contributed to maintaining their livelihood sustainability outcomes (CON). The standardized path coefficients of 0.298 and 0.825 ($p < 0.001$ and $p < 0.01$) under the two models of land transfer out and in, respectively, were the main factors influencing CON. This implies that farmers' participation in land transfer can provide an improvement in the level of family economic welfare and, for each additional unit of land transfer, farmers who transfer into the land receive a larger incremental benefit. Specifically, a farmer's transfer of land use rights adds a fixed monthly income, the "return" from the transfer of use rights. This return gradually decreases over time, reducing the ability to stimulate farmers to improve their livelihoods and making it difficult to achieve stable long-term income growth. In contrast, farmers who expand the scale of agricultural production through land transfer further integrate resources based on their superior productivity and receive larger and more sustainable increments of income, under the influence of scale effects. It can be seen that for farmers with a certain production capacity, a moderate scale will bring a better welfare enhancement effect.

Above all, the existing literature has predominantly focused on the externalities of land transfer [84,85], poverty reduction mechanisms [86–88], and market mechanisms among farmers [89,90]. Wu et al. conducted a study using data from 2011 to 2014 in rural China, which revealed that farmer participation in land transfer led to a significant reduction in fertilizer application intensity in a grain-growing region in Western China [91]. Xie et al. analyzed the legal land transfer rights granted to farmers in rural China from 1999 to 2008 and found that enhancing these rights could promote urban–rural labor migration and reduce income disparities [92]. Tang et al. explored the relationship between agricultural land transfer and carbon emissions using panel data from 30 provincial regions from 2005 to 2019 [93]. This study contributes by focusing on various livelihood assets that influence farmers' decisions regarding land transfer, unlike previous studies that only consider a single asset. This study systematically analyzes the decision-making process of farmers' land transfer behavior based on the sustainable livelihood analysis framework. It identifies the main factors influencing farmers' land transfer decisions, uncovers the mechanisms and transmission paths between different variables, and expands the existing framework of sustainable livelihood analysis. Previous studies have somewhat overlooked environmental/contextual vulnerability and the dimensions of policy, institutions, and processes, limiting a comprehensive understanding of the influences on farmers within this framework. By incorporating a more comprehensive sustainable livelihood analysis framework, this study addresses these gaps and enriches the research content. In terms of research methods, traditional single-factor statistical analyses used in previous studies may not capture the complexity of farmers' land transfer decision-making processes. To overcome this limitation, this study employs structural equation modeling to provide a more nuanced understanding of the decision-making mechanisms across different types of land transfers.

## 4. Conclusions and Policy Implication

This study constructs an analytical framework of "livelihood assets–livelihood strategies–livelihood outcomes" in farmers' land transfer decision-making processes. Based on survey data collected in Sujiatun District, Shenyang City, Donggang City, Dandong City, and Liaoning Province, a structural equation model was used to study the factors and path mechanisms influencing farmers' land transfer behavior in eight dimensions—natural

assets; physical assets (urban); physical assets (rural); human asset; financial assets; social assets; environmental/contextual vulnerability; and policies, institutions, and processes. The main findings of this study are as follows. First, farmers' decisions on land transfer behavior are consistent with the sustainable livelihoods analysis framework and follow the logical paradigm of " environmental/contextual vulnerabilities + policies, institutions or processes + livelihood assets − farmers' livelihood strategies − livelihood outcomes" in general. The outputs of farmers' livelihood strategies can sustain the sustainability of their livelihoods; both livelihood assets and externalities have significant effects on livelihood strategies and outcomes. Second, natural assets (NAT) and external environmental vulnerability (ENV) are important factors influencing farmers' land transfer out and decisions, both of which have a high "disincentive" effect on the formulation of livelihood strategies. Human assets (HUM) are the most basic motivation for farmers to participate in land transfer and thus maintain their livelihood sustainability. In particular, farmers with higher levels of education (hum1) are more likely to engage in off-farm employment compared to large-scale production. In addition, financial assets (FIN), rural physical assets (VIL), and social assets (SOC) all have different degrees of influence on farmers' decision-making, reflecting, to some extent, farmers' risk-averse psychological motivation. Third, human and social assets in the group transferred to land have more significant indirect effects on farmers' livelihood outcomes under the formulation of livelihood strategies. For the group of farmers who transferred their land out, the indirect effect of livelihood assets on sustainable development was weaker and mainly manifested as directness.

Based on the above findings, this study suggests several key policy recommendations. Firstly, it is crucial to enhance the social security system in rural areas to foster a conducive employment environment. Farmers, as key economic actors, often face the decision between expanding their operations or diversifying their activities during the process of agricultural modernization in China. They can either scale up their production through large-scale cultivation or opt for part-time production by transferring their land and engaging in other activities through migration. Secondly, there is a need for the government to focus on enhancing farmers' skills through education and training. The study highlights that human capital plays a fundamental role in enabling farmers to sustain their livelihoods, whether they choose to transfer their land or acquire land use rights from others. Developing high-quality human capital is essential for boosting agricultural productivity, ensuring food security, fostering innovation in agricultural technology, and promoting sustainable agricultural development. Lastly, the government should consider providing targeted subsidies for land transfer in hilly areas. Steeper slopes are more prone to poverty as topographical challenges and limited economic resources of farmers hinder large-scale agricultural operations, while environmental and contextual vulnerabilities further impede land transfer.

This study, despite its potential contribution, has some limitations. Firstly, while it presents a more comprehensive framework for SLA compared to previous studies, it lacks manageability, in terms of time series analysis. The emphasis on a dynamic process in the SLA framework highlights the need for optimization when using 'static' cross-sectional data to study land transfer. Although this is the case, the research results verified the theoretical analysis framework and revealed the impact paths and internal mechanisms of different livelihood assets on farmers' land transfer decisions under the sustainable livelihood analysis framework, which can still have theoretical significance and practical value. The results of this study can still have theoretical significance and practical value. Secondly, although the study incorporates external factors such as policies, institutions, procedures, and environmental vulnerability into the model, to explore the complete path of farmers' land transfer under the SLA framework, the design of indicators still has limitations. Future research could focus on expanding the scope of the study by developing a more scientifically sound indicator system for measuring livelihood capital and incorporating dynamicity analysis through coverage surveys or telephone callbacks to observe key factors over time. This would offer a deeper understanding of the impact of

various factors on farmers' behavior under the SLA framework, enabling more targeted policy suggestions.

**Author Contributions:** Conceptualization, H.L.; methodology, H.Z. and Y.X. (Ying Xue); software, H.Z. and H.L.; validation, H.L. and Y.X. (Ying Xue); formal analysis, Y.X. (Ying Xue) and Y.X. (Yuxuan Xu); investigation, H.Z. and Y.X. (Yuxuan Xu); resources, H.L.; data curation, Y.X. (Ying Xue); writing—original draft preparation, H.Z. writing—review and editing, H.L. and Y.X. (Yuxuan Xu); supervision, H.L.; project administration, H.L.; funding acquisition, H.L. and Y.X. (Ying Xue). All authors have read and agreed to the published version of the manuscript.

**Funding:** This research was funded by the National Natural Science Foundation of China (72074153 and 72103143), the National Key R&D Program Project (2022YFD1901601-1 and 2023YFD15011018), the Liaoning province philosophy and social science young talents training subject commissioned (2022lslqnrcwtkt-51), the Liaoning Province Scientific Research Funding Program (LJKR0239), and the Liaoning Provincial Social Science Planning Fund Project (L22AGL017).

**Data Availability Statement:** The data presented in this study are available on request from the corresponding author. The data are not publicly available due to privacy restrictions.

**Conflicts of Interest:** The authors declare no conflicts of interest. Informed consent was obtained from all individual participants included in the study.

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
