# Peer review of "Decision-Making Mechanism of Farmers in Land Transfer Processes Based on Sustainable Livelihood Analysis Framework: A Study in Rural China"

_land, doi:10.3390/land13050640_

Round 1
Reviewer 1 Report
Comments and Suggestions for Authors
Article review "Decision-Making Mechanism of Farmers in Land Transfer Process based on Sustainable Livelihood Analysis Framework: A Study in Rural China."
The article is even interesting. The Methodology and data seem correct, but the text needs correction.
Comments are in order as they appear while reading the text.
Abstract - The content needs to be more transparent about what the research is about and what the results are.
Introduction - the introduction to the article completely fails to convey what the reader can expect from the content. The same is true of the purpose of the research. According to the authors, "Based on the framework of sustainable livelihood analysis, this study constructs a theoretical analysis framework based on "livelihood assets-livelihood strategies-livelihood outcomes" to clarify the influence of livelihood capital on farmers' land transfer behavior and its mechanism of action...". Unfortunately, the chapter lacks an introduction to the research topic.
The text is repetitive - lines 73-83
Theoretical analysis - As I understand it, the basis of this study was the DFID study and developed through it: the theoretical analysis framework of "livelihood assets-livelihood strategies-livelihood outcomes" of farmers' participation in land transfer under the combined effect of five major livelihood assets, policies, and environmental vulnerabilities - and this is the basis for the mechanisms of influence described in chapters 2.1 and 2.2 ....
And here I wonder about this chapter ... whether it is an introduction to the research methodology and whether it should not be included in the "Introduction" as a theoretical framework and part of the research methodology - such as Figure 1
Materials and methods
Research area - well described.
Data source - Here, I still have doubts; it is not quite clear what and how we will research, and the data has already been described. The choice of samples - towns for analysis may be appropriate, but already the data in Table 2, "Basic characteristics of the sample," is little understood.
There is also the question of whether the responses of 811 people give meaningful results. All in all, it's a lot, but it depends on the scale.
Survey methodology
In general, the Methodology - in the theoretical part is described correctly. But ... "Based on the theoretical framework of "livelihood capital - livelihood strategy - 271 livelihood outcome", this study set up 10 latent variables and 20 measurable variables 272 (Table 3)." - but this needs to be justified and in detail why such and not a different selection of variables, etc. Admittedly, in 3.2, there is a characterization of the selected variables, but before that, there is no explanation regarding their selection. And one can always argue about the validity of the selection of environmental, economic, or social factors - their rank or the way they are measured. In this case, one feels a certain deficiency.
Results.
And so: "Based on Amos 24.0 software, this study conducted exploratory modeling of the sample data using the great likelihood method, and the model was modified in an asymptotic manner to construct the best-fit model. - how was the model fit determined?
Section 4.2. -4.3 - somewhat resembles a discussion
There are also statements about hypothesis verification, which were not mentioned before. My guess is that what is meant is what is recorded in Table 3.
Discussion - the layout seems appropriate; there are positives, criticisms, and plans. However, the discussion of the results obtained is missing here.
Conclusions and recommendations are correct.
In general, the article is interesting. Tex so from the borderline of the social-environmental-economic-spatial topic. Therefore, there is always a risk of negative evaluation of the choice of parameters for the study. But in this case, it can be considered interesting.
However, I recommend that the text be shortened and rewritten and, above all, the purpose of the analysis be clearly described. This one only emerged from the text in chapters 4-5.
Author Response
Dear Editor:
Thank you for giving us a chance to improve the manuscript, entitled “Decision-Making Mechanism of Farmers in Land Transfer Process base on Sustainable Livelihood Analysis Framework: A Study in Rural China” (ID: Land- 2945410). We appreciate the constructive comments from anonymous reviewers, which are very helpful for us revising and improving our paper. We have studied the comments carefully and have made necessary corrections accordingly. We believe the manuscript has significantly improved.
To better show what has been changed, we enclose the manuscript in "Track Changes" mode. In addition, we summarize the point-by-point response as below. Note that the Lines numbers mentioned in the following responses are according to the revised manuscript. And our responses are marked in Blue.
Detailed responses to the reviewer’s comments:
Point 1: Abstract - The content needs to be more transparent about what the research is about and what the results are.
Response 1: We appreciate this comment. We have rewritten the abstract in lines 9-28.
Point 2: Introduction - the introduction to the article completely fails to convey what the reader can expect from the content. The same is true of the purpose of the research. According to the authors, "Based on the framework of sustainable livelihood analysis, this study constructs a theoretical analysis framework based on "livelihood assets-livelihood strategies-livelihood outcomes" to clarify the influence of livelihood capital on farmers' land transfer behavior and its mechanism of action...". Unfortunately, the chapter lacks an introduction to the research topic.
Response 2: Good point. The main purpose is to explain why it is necessary to study the decision-making mechanism of farmers’ land transfer from the perspective of sustainable livelihood. In order to express this meaning more clearly, we add and modify relevant content in lines 49-51, 72-75, 117-121.
Point 3: The text is repetitive - lines 73-83
Response 3: Thanks for this comment. We have revised the text in lines 99-105.
Point 4: Theoretical analysis - As I understand it, the basis of this study was the DFID study and developed through it: the theoretical analysis framework of "livelihood assets-livelihood strategieslivelihood outcomes" of farmers' participation in land transfer under the combined effect of five major livelihood assets, policies, and environmental vulnerabilities - and this is the basis for the mechanisms of influence described in chapters 2.1 and 2.2 .... And here I wonder about this chapter ... whether it is an introduction to the research methodology and whether it should not be included in the "Introduction" as a theoretical framework and part of the research methodology - such as Figure 1 Materials and methods.
Response 4: This is a good point. We added the “theoretical analysis” in “materials and methods” in lines 294-367.
Point 5: Data source - Here, I still have doubts; it is not quite clear what and how we will research, and the data has already been described. The choice of samples - towns for analysis may be appropriate, but already the data in Table 2, "Basic characteristics of the sample," is little understood. There is also the question of whether the responses of 811 people give meaningful results. All in all, it's a lot, but it depends on the scale
Response 5: we have revised the data source to get more clearer in line 251-255, 263,266, 281-282.
Point 6: Survey methodology In general, the Methodology - in the theoretical part is described correctly. But ... "Based on the theoretical framework of "livelihood capital - livelihood strategy - 271 livelihood outcome", this study set up 10 latent variables and 20 measurable variables 272 (Table 3)." - but this needs to be justified and in detail why such and not a different selection of variables, etc. Admittedly, in 3.2, there is a characterization of the selected variables, but before that, there is no explanation regarding their selection. And one can always argue about the validity of the selection of environmental, economic, or social factors - their rank or the way they are measured. In this case, one feels a certain deficiency.
Response 6: We appreciate this comment. We revised the section 2.3.2 in lines 299-404, 445, 448-451.
Point 7: Results. And so: "Based on Amos 24.0 software, this study conducted exploratory modeling of the sample data using the great likelihood method, and the model was modified in an asymptotic manner to construct the best-fit model. - how was the model fit determined?
Response 7: The fit index of the SEM mainly depends on the numerical values ​​of RMESA, GFI, CFI, IFI, TLI. The corresponding values ​​of the model in this study meet the relevant requirements, so it is judged that the suitability of the model meets the requirements.
Table 4. Key indicators for model fitting
|
Type of land transfer |
Name of the index |
Abbr. |
Acceptable Fit Values |
Fit Values |
Results |
|
Land transfer out |
Root Mean Square Error of Approximation |
RMESA |
<0.08 |
0.050 |
Accept |
|
Goodness-of-fit ind |
GFI |
>0.9 |
0.937 |
Accept |
|
|
Comparative fit index |
CFI |
>0.9 |
0.953 |
Accept |
|
|
Incremental fit index |
IFI |
>0.9 |
0.953 |
Accept |
|
|
Tacker–Lewis index |
TLI |
>0.9 |
0.943 |
Accept |
|
|
Cardinality to Degrees of Freedom Ratio |
CMIN/DF |
<3 |
2.445 |
Accept |
|
|
Land transfer in |
Root Mean Square Error of Approximation |
RMESA |
<0.08 |
0.054 |
Accept |
|
Goodness-of-fit ind |
GFI |
>0.9 |
0.931 |
Accept |
|
|
Comparative fit index |
CFI |
>0.9 |
0.949 |
Accept |
|
|
Incremental fit index |
IFI |
>0.9 |
0.949 |
Accept |
|
|
Tacker–Lewis index |
TLI |
>0.9 |
0.938 |
Accept |
|
|
Cardinality to Degrees of Freedom Ratio |
CMIN/DF |
<3 |
2.637 |
Accept |
Point 8: Section 4.2. -4.3 - somewhat resembles a discussion.
Response 8: We agree. We have merged the section 4 and section 5 into 3. Results and Discussion in lines 557,750-774.
Point 9: There are also statements about hypothesis verification, which were not mentioned before. My guess is that what is meant is what is recorded in Table 3.
Response 9: We have deleted the hypothesis verification and the Table 3 in lines 445, 594, 614-615, 637-638, 658, 679-680, 702, 724.
Point 10: Discussion - the layout seems appropriate; there are positives, criticisms, and plans. However, the discussion of the results obtained is missing here.
Response 10: We have merged the section 4 and section 5 into 3. Results and Discussion in lines 557 750-774. The content of criticisms, and plans added to the section 4. Conclusions and Policy Implication in lines 928-946.
Point 11: In general, the article is interesting. Tex so from the borderline of the social-environmental-economic-spatial topic. Therefore, there is always a risk of negative evaluation of the choice of parameters for the study. But in this case, it can be considered interesting. However, I recommend that the text be shortened and rewritten and, above all, the purpose of the analysis be clearly described.
This one only emerged from the text in chapters 4-5.
Response 11: we have shortened the text, the section 2 has been adjusted to the section 2.3.1 according to your suggestion, the section 4 and section 5 have been merged into 3. Results and Discussion. We have rewritten section 4. Conclusions and Policy Implication and the abstract
Thanks for your constructive comments—which helps us a lot to improve the manuscript. We tried our best to address your concerns.

Reviewer 2 Report
Comments and Suggestions for Authors
Thank you very much for this intersting paper. I have only few comments for an improveent of the paper.
1) I would ask the authors to say a bit more about thei theorethical framework and in particular about the "Livelihood assets-livelihood strategies-livelihood outcomes" and figure 1. A bit more relevant literature review on such approach would help the reader who is not familiar with this approach.
2) the paper is about a specific case study, how and to what extent would that be extended as a general strategy for the whole country? if that is nbot the case, then why is that specific case study interseting for the interational readers, apart form the methodological application, which is quite intersting?
3) Information about the data set needs a bit more explaination, especially SOC (social aspects).
4) Limitations of the work are clearly expressed, but the issue of the lack of a dynamic process is quite relevant, and the authors should be more convincng that as it is (static) the analysis of sustainable livelyhood is still valid.
5) the analysis of the literature on similar studies or similar models used should be more effective in framing this work in the currenrt interantional literature.
6) Minor English revisions and editing are needed. See for example section 5.1 (lines 620-626).
Comments on the Quality of English LanguageMinr English revisions and editing are required.
Author Response
Dear Editor:
Thank you for giving us a chance to improve the manuscript, entitled “Decision-Making Mechanism of Farmers in Land Transfer Process base on Sustainable Livelihood Analysis Framework: A Study in Rural China” (ID: Land- 2945410). We appreciate the constructive comments from anonymous reviewers, which are very helpful for us revising and improving our paper. We have studied the comments carefully and have made necessary corrections accordingly. We believe the manuscript has significantly improved.
To better show what has been changed, we enclose the manuscript in "Track Changes" mode. In addition, we summarize the point-by-point response as below. Note that the Lines numbers mentioned in the following responses are according to the revised manuscript. And our responses are marked in Blue.
Detailed responses to the reviewer’s comments:
Point 1: I would ask the authors to say a bit more about their theoretical framework and in particular about the "Livelihood assets-livelihood strategies-livelihood outcomes" and figure 1. A bit more relevant literature review on such approach would help the reader who is not familiar with this approach.
Response 1: Following your suggestion, we have added the content in lines 295-300,311-318.
“The livelihood of farmers has been a topic of concern in various countries, regions, and academic circles [53–56]. Early studies on livelihood primarily centered on poverty, specifically on income levels, consumption capacity, and other factors related to basic living needs [57]. As research has progressed, and with the advancement of poverty alleviation practices and theoretical development, it is recognized that income and consumption are no longer the sole indicators to assess poverty [58].”
“On this basis, in this study, the sustainable livelihoods framework of DFID was slightly adjusted by combining it with the direction and area of land transfer to create a new framework, as illustrated in Figure 2. The paper specifically examines decision-making mechanism of farmers' land transfer behaviour, introducing a new solid line arrow denoting ' livelihood assets → livelihood strategies → livelihood outcomes '. Furthermore, it is noted that environmental vulnerabilities and policies can indirectly influence livelihood strategies through livelihood assets, and may also have a direct impact on farmers' livelihood strategies.”
Point 2: the paper is about a specific case study, how and to what extent would that be extended as a general strategy for the whole country? if that is not the case, then why is that specific case study interseting for the interational readers, apart form the methodological application, which is quite intersting?
Response 2: We appreciate this comment. We have revised the Research area in lines 200-204, and added the content in lines 240-247. At the same time, research by similar scholars in the world has been added.
“Third, the two study areas are located in the core area of black soil protection in Northeast China. The relevant research results have very important practical significance for promoting moderate-scale management of land resources in the black soil area of Northeast China, improving the utilization efficiency of cultivated land re-sources, and realizing the connection between small farmers and modern agriculture. At the same time, it also provides a good reference for international researchers doing relevant research in case analysis. [50-52]”
- Xue, Y.; Xu, Y.; Lyu, J.; Liu, H. The Effect of Uncertainty of Risks on Farmers' Contractual Choice Behavior for Agricultural Productive Services: An Empirical Analysis from the Black Soil in Northeast China. Agronomy-Basel 2022, 12, doi: 10.3390/agronomy12112677.
- Theesfeld, I.; Jelinek, L. A misfit in policy to protect Russia's black soil region. An institutional analytical lens applied to the ban on burning of crop residues. Land Use Policy 2017, 67, 517-526, doi: 10.1016/j.landusepol.2017.06.018.
- Li, W.; Wang, D.; Li, H.; Liu, S. Urbanization-induced site condition changes of peri-urban cultivated land in the black soil region of northeast China. Ecol. Indic. 2017, 80, 215-223, doi: 10.1016/j.ecolind.2017.05.038.
Point 3: Information about the data set needs a bit more explaination, especially SOC (social aspects).
Response 3: We followed the suggestion to add the information in lines 219-222, 227-230.
“According to data from the Sujiatun Bureau of Statistics, the GDP was 22,715.45 million yuan in 2019, the total value of agricultural production accounted for 6.78%; the total population being 425,000, rural population accounted for 14.21%..”
“According to data from Donggang Bureau of Statistics, the total regional production was 21,284.51 million yuan in 2019, the total value of agricultural production ac-counted for 35.78%; the total population was 592,248, rural population was accounted for 73.78%.”
Point 4: Limitations of the work are clearly expressed, but the issue of the lack of a dynamic process is quite relevant, and the authors should be more convincng that as it is (static) the analysis of sustainable livelyhood is still valid.
Response 4: We agree. We have revised the limitations of the work in lines 933-937.
“Although this is the case, The research results verified the theoretical analysis framework and revealed the impact paths and internal mechanisms of different livelihood assets on farmers' land transfer decisions under the sustainable livelihood analysis framework, which can still have theoretical significance and practical value.”
Point 5: the analysis of the literature on similar studies or similar models used should be more effective in framing this work in the currenrt interantional literature.
Response5: Thanks for the suggestion. We have added the currenrt interantional literature throughout the manuscript in line 1157-1373.
- Akimowicz, M.; Landman, K.; Kephaliacos, C.; Cummings, H. Toward Agricultural Intersectionality? Farm Intergenerational Transfer at the Fringe. A Comparative Analysis of the Urban-Influenced Ontario's Greenbelt, Canada and Toulouse InterSCoT, France. Front. Sustain. Food Syst. 2022, 5, doi: 10.3389/fsufs.2021.759638.
- Huysentruyt, M.; Barrett, C.B.; Mcpeak, J.G. Understanding Declining Mobility and Inter-household Transfers among East African Pastoralists. Economica 2009, 76, 315-336, doi: 10.1111/j.1468-0335.2007.00675.x.
- Niu, K.; Xu, H. Does urban-rural integration reduce rural poverty ? Agribusiness 2024, doi: 10.1002/agr.21935.
- Urfels, A.; Mausch, K.; Harris, D.; Mcdonald, A.J.; Kishore, A.; Balwinder-Singh; van Halsema, G.; Struik, P.C.; Craufurd, P.; Foster, T.; et al. Farm size limits agriculture's poverty reduction potential in Eastern India even with irrigation-led intensification. Agric. Syst. 2023, 207, doi: 10.1016/j.agsy.2023.103618.
- Bird, K.; Chabe-Ferret, B.; Simons, A. Linking human capabilities with livelihood strategies to speed poverty reduction : Evidence from Rwanda. World Dev. 2022, 151, doi: 10.1016/j.worlddev.2021.105728.
- Reilly, A.; Van Rooy, D.; Angus, S. An Exploration of the Relationship Between Personality and Strategy Formation Using Market Farmer : Using a Bespoke Computer Game in Behavioural Research. Entertainment Computing and Serious Games, Icec-Jcsg2019 2019, 11863, 311-323, doi: 10.1007/978-3-030-34644-7_25.
- Assassi, S.; Daoudi, A.; Amokrane, A. The land market as a mechanism for reallocating water in the irrigated scheme of Guelma, Algeria. Cah. Agric. 2022, 31, doi: 10.1051/cagri/2022028.
Point 6: Minor English revisions and editing are needed. See for example section 5.1 (lines 620-626).
Response 6: Good points. We have rewritten the section 5.1 in lines 750-774. The manuscript has been re-polished in English throughout the manuscript.
Special thanks to you for your good comments. We tried our best to improve the manuscript and made some changes in the manuscript. It is hoped that the correction will meet with approval.
Once again, thank you very much for your comments and suggestions.

Round 2
Reviewer 1 Report
Comments and Suggestions for Authors
After the correction, the article has gained in value. As I have already noted: The text so from the borderline of the social-environmental-economic-spatial topic. Therefore, there is always a risk of negative evaluation of the choice of parameters for the study. But in this case, it can be considered interesting. But in its current form, I recommend it for publication.